# T cell responses at diagnosis of amyotrophic lateral sclerosis predict disease progression

Solmaz Yazdani[1,6], Christina Seitz[1,6], Can Cui[1,6], Anikó Lovik[1], Lu Pan [2], Fredrik Piehl [3,4], Yudi Pawitan [2], Ulf Kläppe [3,4], Rayomand Press[3,4], Kristin Samuelsson[3,4], Li Yin[2], Trung Nghia Vu [2], Anne-Laure Joly[3], Lisa S. Westerberg [5], Björn Evertsson[3,4], Caroline Ingre[3,4,6], John Andersson [1,6] ✉ & Fang Fang [1,6] ✉

Amyotrophic lateral sclerosis (ALS) is a fatal neurodegenerative disease, involving neuroinflammation and T cell infiltration in the central nervous system. However, the contribution of T cell responses to the pathology of the disease is not fully understood. Here we show, by flow cytometric analysis of blood and cerebrospinal fluid (CSF) samples of a cohort of 89 newly diagnosed ALS patients in Stockholm, Sweden, that T cell phenotypes at the time of diagnosis are good predictors of disease outcome. High frequency of CD4$^+$FOXP3$^-$ effector T cells in blood and CSF is associated with poor survival, whereas high frequency of activated regulatory T (Treg) cells and high ratio between activated and resting Treg cells in blood are associated with better survival. Besides survival, phenotypic profiling of T cells could also predict disease progression rate. Single cell transcriptomics analysis of CSF samples shows clonally expanded CD4$^+$ and CD8$^+$ T cells in CSF, with characteristic gene expression patterns. In summary, T cell responses associate with and likely contribute to disease progression in ALS, supporting modulation of adaptive immunity as a viable therapeutic option.

Amyotrophic lateral sclerosis (ALS) is characterized by selective degeneration of upper and lower motor neurons in the brain cortex and spinal cord, leading eventually to paralysis of respiratory muscles and death[1]. The pathogenesis of ALS remains poorly understood, but neuroinflammatory features are hallmarks of neurodegenerative diseases, including ALS[2,3]. The presence of lymphocytes in the spinal cord and areas of neuronal injury has been suggested to implicate a role in adaptive immune responses[4], where T cells constitute the main lymphocyte subtype infiltrating the central nervous system (CNS) of ALS patients[5,6]. In addition, infiltration of CD4$^+$ T cells was further shown to be associated with increased levels of CCL2 and activation of microglia[7].

Experimental studies utilizing *SOD1* mice that overexpress mutant superoxide dismutase 1 and develop ALS-like disease support the role

of T cells in motor neuron disease[8,9]. *SOD1* mice bred onto a T cell-receptor (TCRβ) deficient background, leading to loss of T cells, displayed accelerated disease progression than controls, suggesting a neuroprotective role of this cell type[10]. In an analogous manner, *SOD1* mice lacking the *RAG2* gene with the absence of both B and T cells displayed accelerated disease progression[11]. B cell deficiency by itself does, however, not exert a noticeable impact on the disease characteristics of *SOD1* mice[12].

The potential neuroprotective role of T cells has been further studied in clinical materials, providing indications of a protective role of CD4$^+$FOXP3$^+$ regulatory T (Treg) cells, as the number of Treg cells in the blood of patients with ALS correlated negatively with a rate of disease progression[9,13]. Indeed, the adoptive transfer of Treg cells into

[1]Institute of Environmental Medicine, Karolinska Institutet, Stockholm, Sweden. [2]Department of Medical Epidemiology and Biostatistics, Karolinska Institutet, Stockholm, Sweden. [3]Department of Clinical Neuroscience, Karolinska Institutet, Stockholm, Sweden. [4]Neurology clinic, Karolinska University Hospital, Stockholm, Sweden. [5]Department of Microbiology, Tumor, and Cell Biology, Karolinska Institutet, Stockholm, Sweden. [6]These authors contributed equally: Solmaz Yazdani, Christina Seitz, Can Cui, Caroline Ingre, John Andersson, Fang Fang. ✉e-mail: john.andersson@ki.se; fang.fang@ki.se

*SOD1* mice slowed down disease progression and extended the survival of recipient mice[8]. In addition, *SOD1* mice treated with a complex of interleukin-2 (IL-2) and anti-IL-2 antibodies to enhance endogenous Treg cell populations displayed slower disease progression and longer survival than untreated controls[9]. Clinical trials involving therapeutic targeting of T cells in ALS are still rare. However, in the phase 2 randomized trial, the administration of low-dose interleukin-2 (ld-IL-2) was shown to be well tolerated and increase the percentage of Treg cells[14]. In another study, infusion of expanded autologous Tregs was found to lead to an increasing percentage and suppressive function of Tregs and slowing of the disease progression[2].

Here we show, through flow cytometric analysis of blood and cerebrospinal fluid (CSF) samples of a cohort of 89 newly diagnosed ALS patients in Stockholm, Sweden, that a high proportion of CD4+ FOXP3− effector T cells is associated with poor survival, whereas a high proportion of activated Treg cells is beneficial. In addition, the composition of T cell subsets predicts disease progression. Finally, single-cell RNA sequencing demonstrates the presence of clonally expanding CD4+ cytotoxic lymphocytes in ALS patients but not controls. These findings add new evidence to the involvement of T cells in the disease progression of ALS and support the modulation of adaptive immunity as a potential therapeutic option for the disease.

## Results

We included in the analysis a cohort of 89 patients with newly diagnosed ALS between March 2016 and March 2020, who were individually followed from the date of diagnosis until the occurrence of death (or use of invasive ventilation) or October 7, 2020, at the ALS Clinical Research Centre of Karolinska University Hospital in Stockholm, Sweden. The clinical characteristics were comparable between these 89 patients and all newly diagnosed ALS patients during the study period in Stockholm (i.e., source population, $N = 245$), suggesting good generalizability of results (Table 1).

**Table 1 | Characteristics of the study cohort and the entire population of ALS patients diagnosed during the study period in Stockholm, according to the Swedish Motor Neuron Disease Quality Registry**

| Characteristics | Patients of the study cohort (n = 89) | All ALS patients (n = 245) |
|---|---|---|
| **Age at diagnosis, years** | | |
| Mean (SD) | 66.52 (10.69) | 66.16 (11.18) |
| **Sex, N (%)** | | |
| Male | 54 (60.67%) | 131 (53.47%) |
| Female | 35 (39.33%) | 114 (46.53%) |
| **Final diagnosis, *N* (%)** | | |
| ALS | 82 (92.13%) | 218 (88.98%) |
| Other MND | 7 (7.87%) | 27 (11.02%) |
| **Site of onset, *N* (%)** | | |
| Bulbar | 38 (42.70%) | 82 (33.88%) |
| Non-bulbar | 51 (57.30%) | 157 (64.88%) |
| Other | — | 3 (1.24%) |
| ALSFRS-R score at diagnosis, mean (SD) | 38.29 (7.85) | 36.84 (8.21) |
| Progression rate at diagnosis, mean (SD) | 0.81 (0.82) | 0.77 (0.70) |
| Diagnostic delay, median | 382 days | 395 days |
| **Survival status at the end of follow-up, *N* (%)** | | |
| Dead | 50 (56.18%) | 132 (53.88%) |
| Alive | 39 (43.82%) | 113 (46.12%) |

## CD4+FOXP3− effector T cells and activated Treg cells are differentially associated with ALS survival

Flow cytometry was used to define T cell subsets in blood and CSF samples collected from each patient at the time of diagnosis, to identify immune markers associated with ALS survival. Supplementary Fig. 1 shows the gating strategy and summary of primary data for T cell subsets. Per standard deviation (SD) increase, the frequency of CD4+ and CD4+FOXP3− effector T cells in the blood at the time of ALS diagnosis was associated with a higher risk of death (Supplementary Table 1). A similar trend was noted in CSF, although not statistically significant. In blood, an increasing activated Treg (aTreg) to resting Treg (rTreg) ratio was associated with a lower risk of death. The exclusion of patients with ALS spectrum disorders, including primary lateral sclerosis, primary spinal muscular atrophy, and other motor neuron disease ($N = 7$) (Supplementary Table 2) or patients with *C9orf72* mutations ($N = 10$) (Supplementary Table 3) only changed these results marginally. These results tended to be stronger among females, older patients, patients with a faster progression rate at the time of diagnosis, and patients with bulbar onset (Table 2).

Studying the frequencies of T cell subsets as categorical variables rendered a similar result pattern (Table 3). Namely, a higher level of CD3+, CD4+, and CD4+FOXP3− effector T cells in either blood or CSF was associated with a higher risk of death, whereas a higher level of aTreg cells in blood was associated with a lower risk of death, after ALS diagnosis.

Although the entire study cohort included 89 patients, not all patients were included in all analyses, due to missing data on covariables (i.e., BMI and ALSFRS-R score) or T cell subsets (i.e., lack of specimen or unsuccessful flow cytometric analysis). A sensitivity analysis including only patients with complete data ($N = 63$) showed similar patient characteristics (Supplementary Table 4) and results (Supplementary Table 5).

## CD4+FOXP3− effector T cells and Treg cells are differentially associated with ALS progression rate

Clinical biomarkers are essential for developing new therapeutic strategies to combat ALS. Therefore, we proceeded to determine if the proportion of any cell population, at the time of diagnosis, was associated with the rate of disease progression. ALS functional rating scale-revised (ALSFRS-R) declined continuously with time since diagnosis, with a decline rate of 0.88 points per month (Supplementary Fig. 2). Patients with higher levels of aTreg and aTreg/rTreg ratio, or lower levels of rTreg in blood had a slower declining rate of ALSFRS-R, compared with other patients (Fig. 1A). The associations of T cell subsets with disease progression rate were stronger in CSF than in blood (all *p* values for difference <0.0001; Fig. 1B). On the other hand, patients with lower levels of CD3+, CD4+, CD8+, or CD4+FOXP3− effector T cells as well as those with higher levels of Treg cells in CSF all displayed a lower rate of declining in ALSFRS-R, compared with patients with the opposite cellular patterns (Fig. 1A, B).

## T cell composition in blood partially reflects T cell composition in CSF

Previous studies on immunity in ALS have mainly focused on the composition of leukocyte subsets in blood with the assumption that it reflects the immunity at the site of the disease. Considering our finding that T cell subsets measured in blood and CSF had different associations with ALS progression rate, we moved on to determine to what extent the composition of T cells measured in blood predicts the composition of T cells in CSF. In the present study, all measured T cell subsets were positively correlated between blood and CSF (Supplementary Table 6). The fact that the Spearman correlation coefficient ranged from 0.20 to 0.33 suggests, however, that the immune profile measured in blood only partially predicts the immune profile in the intrathecal milieu.

**Table 2 | Hazard ratios (HR) and their 95% confidence intervals (CI) of death in relation to per standard deviation increase in the frequency of T cell subsets in blood or cerebrospinal fluid, adjusted for age at diagnosis, sex, site of onset, delay in diagnosis, disease progression rate at diagnosis, body mass index (BMI), and the time difference between measurement of BMI and blood sampling, stratified analysis by sex, age, progression rate, and site of onset**

| Cell population out of live cells (%) | No. of patients (deaths) | HR (95% CI) | No. of patients (deaths) | HR (95% CI) |
|---|---|---|---|---|
| **Sex** | **Female** | | **Male** | |
| CD4$^+$ | 30 (18) | 1.91 (0.94–3.88) | 48 (26) | 1.53 (0.96–2.45) |
| T$_{eff}$ | 29 (18) | **2.17 (1.02–4.63)** | 48 (26) | 1.53 (0.96–2.43) |
| aTreg/rTreg (ratio) | 29 (18) | **0.28 (0.11–0.71)** | 47 (26) | 0.77 (0.47–1.25) |
| **Age** | **Younger than 67** | | **Older than 67** | |
| CD4$^+$ | 39 (18) | 1.09 (0.67–1.79) | 39 (26) | **2.32 (1.31–4.09)** |
| T$_{eff}$ | 39 (19) | 1.12 (0.67–1.87) | 38 (25) | **2.43 (1.33–4.43)** |
| aTreg/rTreg (ratio) | 38 (18) | 0.90 (0.55–1.49) | 38 (26) | **0.47 (0.26–0.85)** |
| **Progression rate** | **Slow (<0.45)** | | **Fast (>=0.45)** | |
| CD4$^+$ | 39 (16) | 1.75 (0.94–3.27) | 39 (28) | **1.76 (1.07–2.88)** |
| T$_{eff}$ | 39 (17) | 1.77 (0.99–3.18) | 38 (27) | **2.05 (1.16–3.64)** |
| aTreg/rTreg (ratio) | 38 (17) | 0.45 (0.15–1.30) | 38 (27) | 0.76 (0.44–1.31) |
| **Site of onset** | **Spinal** | | **Bulbar** | |
| CD4$^+$ | 43 (20) | 1.45 (0.78–2.69) | 33 (23) | **2.00 (1.18–3.39)** |
| T$_{eff}$ | 42 (20) | 1.42 (0.79–2.56) | 33 (23) | **2.14 (1.23–3.71)** |
| aTreg/rTreg (ratio) | 41 (20) | 0.89 (0.48–1.65) | 33 (23) | **0.42 (0.19–0.90)** |

HRs that are statistically significant ($p < 0.05$) are shown in bold format.

## T cell profile in blood predicts disease progression

In addition to studying the T cell subsets individually, we also used exploratory factor analysis (EFA) to understand the synergistic effort of different cell types. In this analysis, we identified five factors that explained 89.5% of the total variance of all variables included in the EFA. These factors correspond to (1) cell proliferation, (2) rTreg, (3) total Treg and aTreg, (4) effector T cells, and (5) FOXP3 expression levels (Fig. 2A). Our analysis showed that 10% increase in the score of Factor (2) (rTreg) was associated with 14% higher risk of death (HR = 1.14, 95% CI = [1.02, 1.27]), whereas Factors (1) and (5) showed statistically significant associations with the declining rate of ALSFRS-R after diagnosis (Fig. 2B). We continued to assess whether the T cell subsets could help to stratify unique patient groups, using cluster analysis. Four clusters of patients (pseudo-$R^2$ = 0.4) were identified in this analysis (Fig. 2C), where Clusters A and D differed significantly from Clusters B and C with regards to Factor 2 and disease progression rate (Fig. 2D). Consequently, the composite T cell profile in blood predicts the disease progression in ALS.

## Altered cell composition and gene expression in CSF of ALS patients

The flow cytometric analysis of CSF from ALS patients revealed that, although T cells constituted the most abundant population of cells in CSF, there were other leukocytes present. We, therefore, performed 5′ scRNA-seq using the 10x Genomics platform on CSF cells collected from five additional ALS patients and four controls (i.e., two patients with normal pressure hydrocephalus, NPH, and two non-inflammatory controls, including one patient with cervical radiculopathy and one healthy control). After manual annotation (Supplementary Fig. 3), the t-SNE analysis revealed distinct cell clusters, including sizable populations of T cell subsets, natural killer (NK) cells, monocytes, and dendritic cells (Fig. 3A). CSF cells of ALS patients displayed increased amounts of CD4$^+$ cytotoxic lymphocytes (CTL) and CD4$^+$ T cells with an activated phenotype (CCR7$^-$CCL5$^+$)[15] but decreased amounts of monocytes and CD4$^+$CD8$^+$ double-positive T cells, compared with controls (Fig. 3A, B). Focusing on genes differentially expressed in the T cell subsets of ALS patients and controls, we found a set of genes that

control for cellular cytotoxicity, including *GNLY*, *GZMA*, *GZMB*, *GZMH*, *GZMK*, *PRF1*, *CTSW*, *KLRB*, *KLRD1*, and *NKG7* (Fig. 3C).

## ALS is characterized by T cell expansions in CSF

In the flow cytometric analysis, we found that high levels of effector T cells were associated with poor survival among ALS patients, which might imply that these cells recognize antigens, expand, and exert effector functions. We used V(D)J TCR repertoire sequencing to study the CSF cells of ALS patients, compared to controls. The presence of multiple identical TCR sequences was considered a sign of T cell expansion. ALS patients exhibited greater levels of TCR expansions in CD4$^+$ CTLs as well as in CD4$^+$ and CD8$^+$ T cells with an activated phenotype (CCR7$^-$CCL5$^+$), compared with controls (Fig. 4A, B). The controls displayed substantial TCR expansions among CD8$^+$ cells, mainly attributable to the two patients with NPH. Nonetheless, T cell expansion in CSF appears to be a feature of ALS, strengthening the view that effector T cells might direct a deleterious immune response in ALS.

## Expanded T cells in ALS display a distinct set of lineage-specification factors

Previous studies have suggested that neuroprotective Th2 cells dominate the early phase of ALS progression that is eventually overtaken by a deleterious Th1 response[16]. To investigate the presence of different CD4$^+$ T cell subsets, we examined GATA3, Eomesodermin (Eomes), Tbet (Tbx21), and RORγt (RORC) expression in the single-cell transcriptomics data. In general, ALS patients, non-inflammatory controls, and NPH control expressed comparable amounts of these lineage-defining transcription factors (Fig. 5A). We then compared the usage of these lineage-defining factors between expanded T cells (defined as >5 identical TCR sequences) and non-expanded T cells (defined as ≤5 identical TCR sequences) among the ALS patients. Expanded T cells had a substantially increased proportion of Eomesodermin expressing cells and a modestly increased proportion of Tbet and GATA3 expressing cells (Fig. 5B). Thus, expanded T cells display a distinct set of lineage-defining transcription factors, compared with non-expanded T cells, in ALS.

**Table 3 | Hazard ratios (HR) and their 95% confidence intervals (CI) of death (or use of invasive ventilation) in relation to the frequency of T cell subsets in blood or cerebrospinal fluid (CSF), after adjustment for age at diagnosis, sex, site of onset, diagnostic delay, disease progression rate at diagnosis, body mass index (BMI), and the time difference between measurement of BMI and blood sampling**

| Cell population out of live cells (%) | Tertile | Blood | | CSF | |
|---|---|---|---|---|---|
| | | No. of patients (outcomes) | HR (95% CI) | No. of patients (outcomes) | HR (95% CI) |
| CD3+ | Low | 26 (12) | Ref | 27 (15) | Ref |
| | Medium | 26 (16) | **2.47 (1.11–5.52)** | 27 (17) | **2.21 (1.01–4.82)** |
| | High | 26 (16) | **2.49 (1.03–6.03)** | 26 (14) | 1.24 (0.54–2.82) |
| CD4+ | Low | 26 (13) | Ref | 27 (14) | Ref |
| | Medium | 26 (15) | 1.82 (0.81–4.06) | 27 (16) | 1.79 (0.82–3.93) |
| | High | 26 (16) | **2.29 (1.04–5.04)** | 26 (16) | **3.04 (1.24–7.46)** |
| CD8+ | Low | 22 (13) | Ref | 23 (10) | Ref |
| | Medium | 22 (13) | 1.04 (0.44–2.46) | 22 (16) | 1.22 (0.47–3.15) |
| | High | 22 (09) | 0.81 (0.32–2.06) | 22 (11) | 0.78 (0.30–2.06) |
| Teff | Low | 26 (13) | Ref | 27 (14) | Ref |
| | Medium | 26 (14) | 1.61 (0.72–3.62) | 26 (15) | 1.68 (0.75–3.77) |
| | High | 25 (17) | **2.43 (1.10–5.37)** | 26 (17) | **3.18 (1.33–7.64)** |
| Treg | Low | 26 (12) | Ref | 27 (17) | Ref |
| | Medium | 26 (14) | 1.44 (0.64–3.23) | 26 (13) | 0.96 (0.44–2.08) |
| | High | 25 (18) | 1.39 (0.64–3.00) | 26 (16) | 0.87 (0.41–1.86) |
| aTreg | Low | 26 (15) | Ref | | |
| | Medium | 25 (12) | **0.40 (0.17–0.92)** | | |
| | High | 25 (17) | 0.76 (0.37–1.59) | | |
| rTreg | Low | 26 (13) | Ref | | |
| | Medium | 26 (16) | 1.65 (0.76–3.60) | | |
| | High | 25 (15) | 1.59 (0.69–3.66) | | |
| aTreg/rTreg (ratio) | Low | 26 (14) | Ref | | |
| | Medium | 25 (16) | 1.06 (0.46–2.43) | | |
| | High | 25 (14) | 0.56 (0.24–1.31) | | |

HRs that are statistically significant ($p < 0.05$) are shown in bold format.

## Discussion

The present study is, to our knowledge, the first to simultaneously define the immunophenotype in both the peripheral and intrathecal compartment at the time of ALS diagnosis and to relate such phenotype to disease progression. We found that a high frequency of effector T cells was indicative of poor survival, whereas a high frequency of activated Treg (or a high ratio of activated to resting Treg) cells was indicative of better survival, after ALS diagnosis. T cell subsets measured in both blood and CSF were also predictive of disease progression rate after ALS diagnosis. Finally, CSF T cells exhibited differential gene expression and clonal expansion patterns between ALS patients and individuals free of ALS.

In the primary analysis, we found that the proportion of CD4+ T cells out of total live cells in blood showed an inverse association with the survival of ALS patients, where the deleterious effect was primarily driven by CD4+ effector T cells. In contrast to previous studies, we did not observe a strong association between total Treg cells and risk of death[8,9,17]. However, we found that a high frequency of aTreg cells or a high ratio between aTreg and rTreg cells was associated with a lower risk of death. These findings extend the previous knowledge by showing that the noted protective role of Treg cells on ALS survival is likely attributable to Treg cell activation. A similar relation was evident also in the intrathecal compartment, although we could not study aTreg cells separately as the cells

present in CSF had previously been activated and lacked CD45RA expression. The absence of a clearly defined rTreg cell population in CSF also made it technically challenging to gate for aTreg cells in a reproducible manner. In sum, our data suggest a harmful role of effector T cells whereas a beneficial role of primarily activated Treg cells in ALS survival. The dual role of T cell immunity in ALS may explain partly the failure of clinical trials using immunosuppressive drugs to treat ALS, as they may inhibit both the beneficial and harmful effects exerted by T cells[18].

In blood, effector T cells were not indicative of disease progression rate, whereas a high frequency of aTreg, high ratio of aTreg/rTreg, and low frequency of rTreg were all indicative of slower declining of ALSFRS-R (i.e., lower disease progression rate). In contrast to blood, we found a strong correlation between a high frequency of effector T cells in CSF and rapid disease progression. Although we could not identify aTreg cells in CSF, a high frequency of total Treg cells in CSF was indeed associated with a slower disease progression. Using factor analysis combined with cluster analysis, we observed that composite T cell profiles in blood were also able to predict the subsequent progression rate of ALS. Altogether, our observations from the different analyses mutually strengthen the notion that distinct T cell phenotypes predict disease progression after ALS diagnosis. This may not only be valuable for deciding how to manage patient care but also for understanding disease diversity in clinical studies.

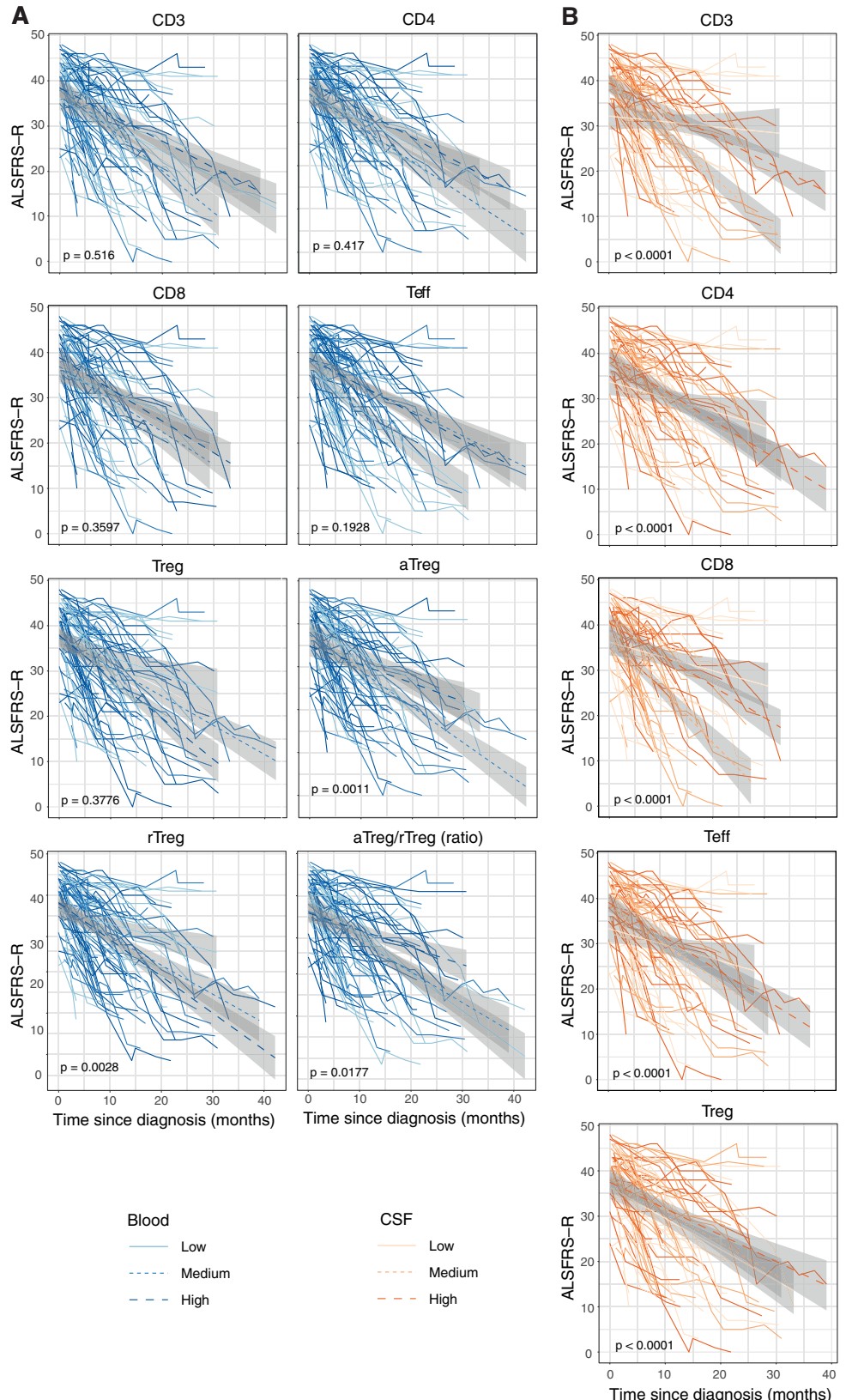

**Fig. 1 | Progression rate of newly diagnosed ALS patients.** ALS patients were stratified by the proportion of T cell subsets in the blood (**A**) and cerebrospinal fluid (**B**) at the time of diagnosis. The longitudinal evolution of ALS functional rating scale-revised (ALSFRS-R) in relation to the baseline categories of T cell subsets is plotted with 95% confidence intervals. The *p* value is of the interaction term of time and T cell subset category from the linear mixed model fitted without adjustments.

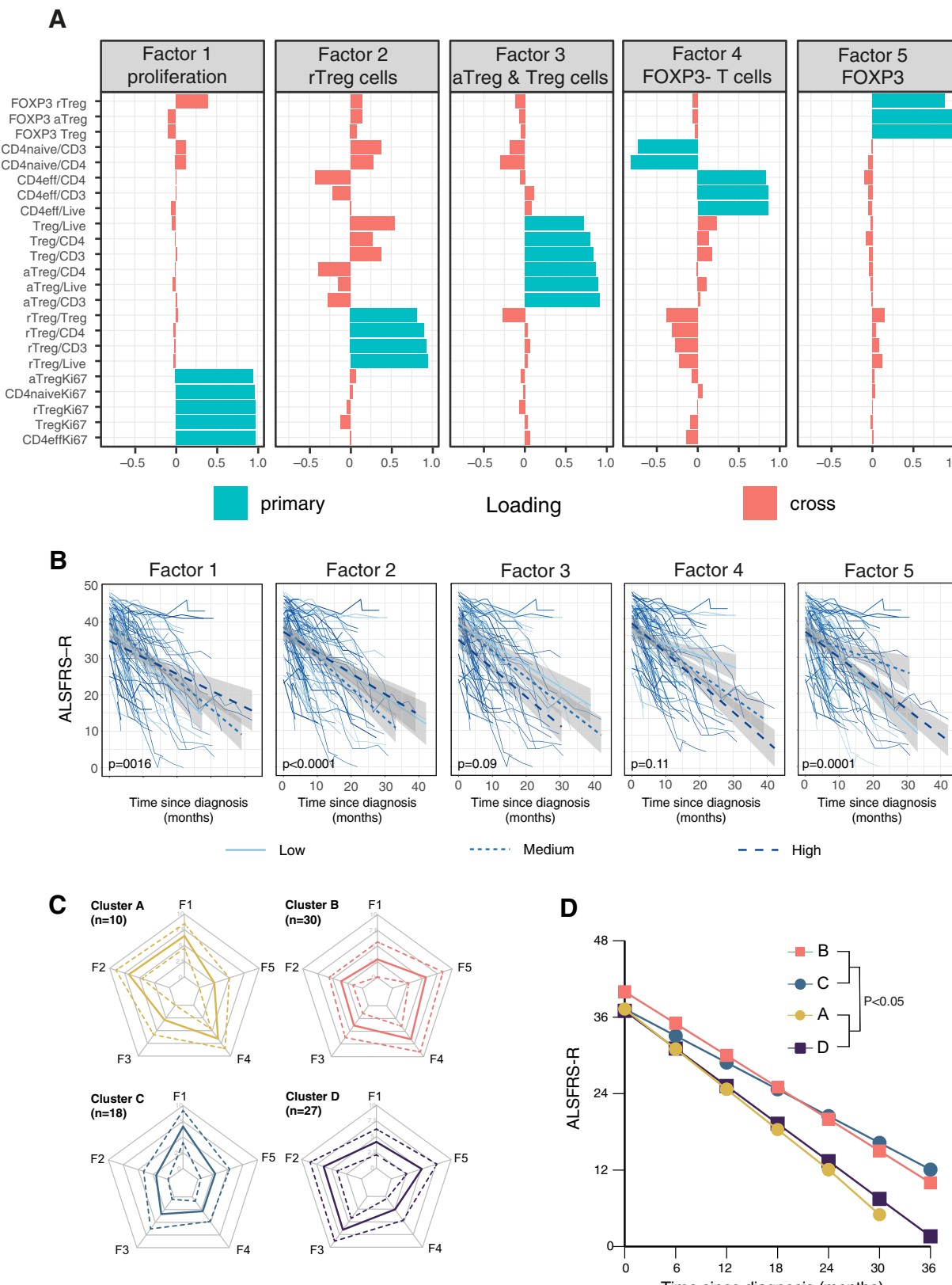

Fig. 2 | Distinct T cell profiles are associated with differential ALS disease progression. Exploratory factor analysis was used to reduce the flow cytometric data into summary variables and resulted in five factors that explained 89.5% of the total variance (A). ALS patients were stratified by the factors. Longitudinal changes in the ALS functional rating scale-revised (ALSFRS-R) in blood were analysed using linear mixed models with mean and 95% confidence intervals for each category of factors. B Cluster analysis was used to identify patients with similar factor profiles and resulted in four clusters. Standardized and centered parallel profile plots of the individual patients' factor scores are colored according to cluster membership (C). Disease progression in the different clusters was analysed using a linear mixed model (D).

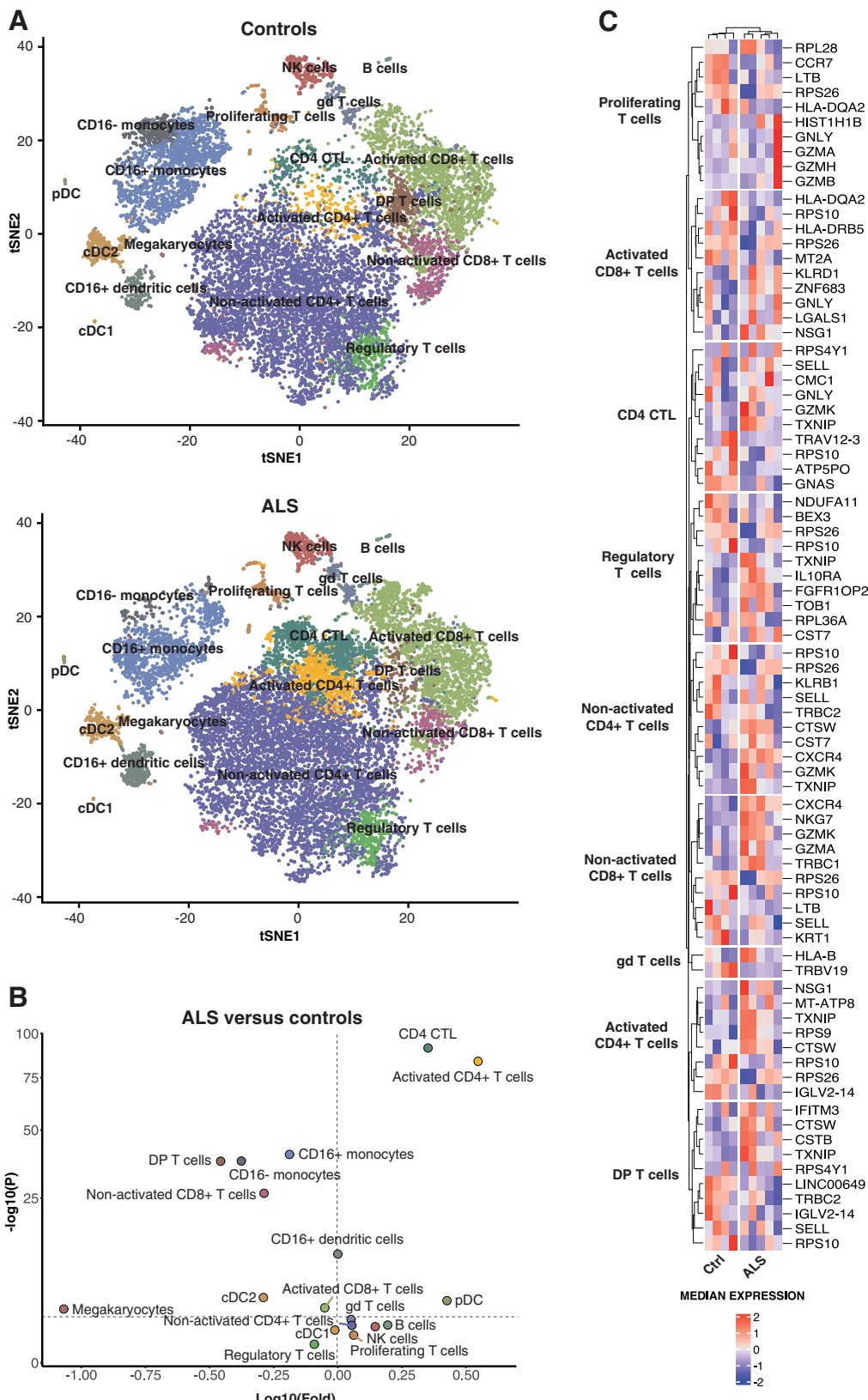

**Fig. 3 | Differential cell composition and gene expression in CSF of ALS patients versus controls.** t-SNE plots of 10X scRNA-seq data showing leukocyte subsets from five ALS patients and four controls (**A**). A pairwise comparison of ALS and controls in terms of cell count abundance for every cell type was performed using a two-sided proportions *z*-test (horizontal dotted lines correspond to *P* = 0.05) which

was followed by Yate's continuity correction. Correction for multiple testing was done using Benjamini–Hochberg (BH) correction (**B**). Heat map illustrating the top-up to five most up- and down-regulated genes in different T cell subsets in ALS patients (*N* = 5) versus controls (*N* = 4) (**C**).

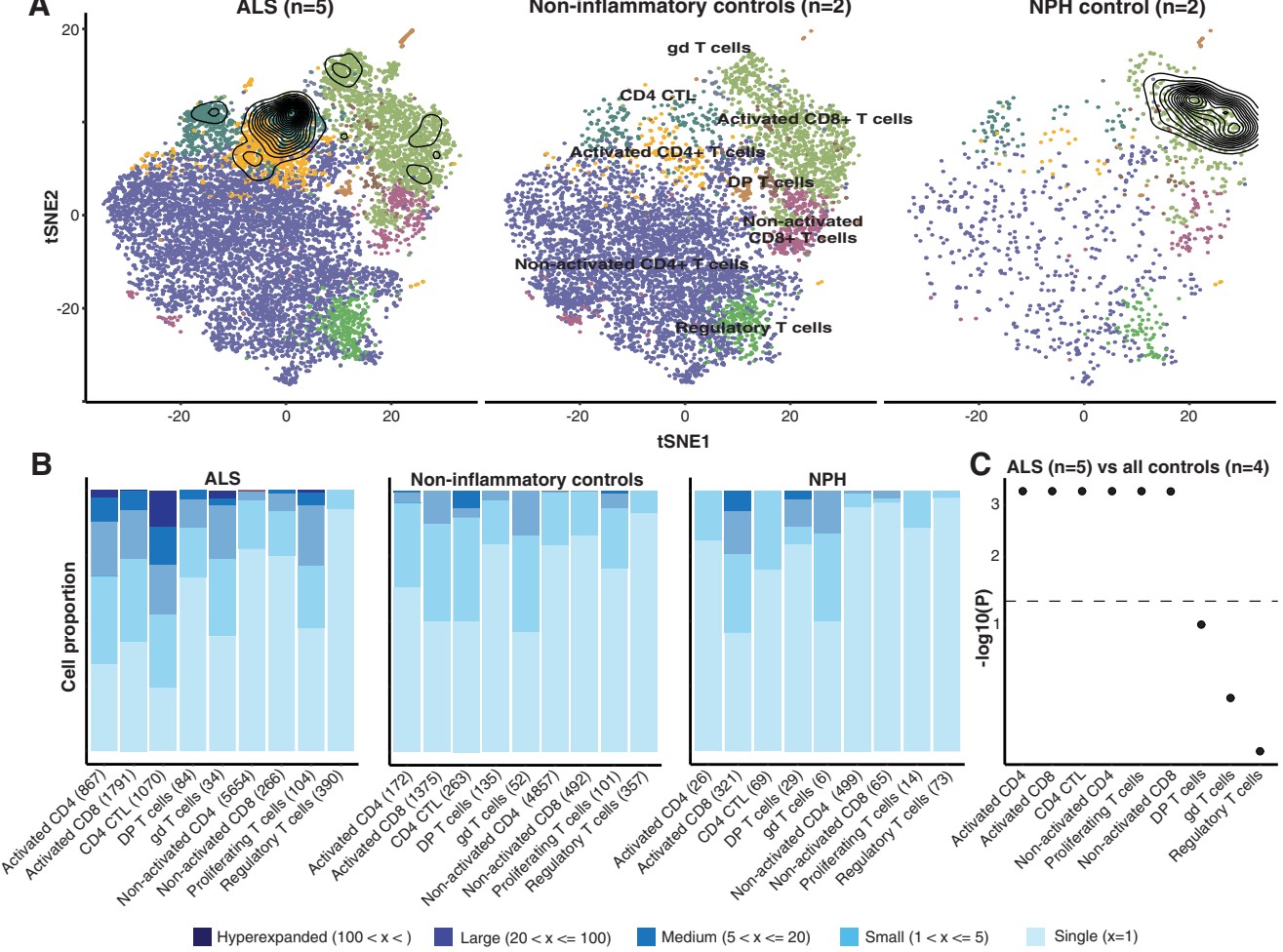

**Fig. 4 | Clonal expansion of CD4⁺ T cell subsets in ALS.** t-SNE plots of 10X scRNA-seq data showing T cell subsets and contour plots outlining TCR expansions identified using 10X VDJ scRNA-seq in five ALS patients, two non-inflammatory controls, and two normal pressure hydrocephalus (NPH) controls individuals (**A**). Quantification of TCR expansion from 10X VDJ scRNA-seq from ALS patients and controls (**B**) with significance displayed in ascending order (**C**). For each cell type, *p* values were calculated using Pearson's Chi-squared test and Monte Carlo simulation with 2000 replicates.

One striking finding was the presence of expanded CD4⁺ and CD8⁺ effector T cell clones in the CSF of ALS patients. Not only does this imply that there are ongoing T cell immune responses in CSF, but it opens the possibility that ALS has an autoimmune component or is driven by a chronic infection[19,20]. Importantly, these expansions occur in FOXP3⁻ effector T cells, rather than in Treg cells, which agrees with our finding that high levels of effector T cells were associated with a poor prognosis after ALS diagnosis. Although Treg cells do not display similar TCR expansion, this does not preclude Treg cells from acting protectively. We and others have previously demonstrated that Treg cells display high levels of TCR ζ-chain phosphorylation allowing them to exert effector functions in response to IL-2 or IL-4 in the absence of TCR stimulation[21,22]. Notably, a surprisingly large proportion of TCRs was found in 2–5 copies among both ALS patients and controls. We speculate that this may reflect a degree of homeostatic proliferation instead of antigen-driven expansion within the intrathecal compartment, which is partially closed off from the periphery by the blood–brain barrier. The observation of clonal expansion does, however, need to be replicated in independent studies and we are currently making efforts to identify the specificity of expanded T cells, which may help unravel the etiopathogenesis of ALS.

Both clinical and animal studies suggest that ALS is initially characterized by a neuroprotective Th2 response that eventually is replaced by a pro-inflammatory Th1/Th17 response[16,23]. The transcriptional circuitry controlling Th1/Th2 differentiation normally promotes lineage stability by being self-reinforcing[24]. The present study may offer a mechanistic insight as to how the shift from Th2 to Th1 occurs. We propose that the strength and nature of TCR recognition during ALS facilitate the differentiation of antigen-specific Eomesodermin expressing CD4⁺ CTLs and Th1 cells. This is in line with the finding that GATA3 dominated among lineage-specification factors in CD4⁺ T cells in the CSF samples of ALS patients collected at the time of diagnosis. Tbet and Eomesodermin were found at a higher proportion in expanded T cells, compared with un-expanded T cells, which presumably will accumulate over time and alter the T cell subset balance over time. Eomesodermin expression in human CD4⁺ T cells drives the production of the Th1 cytokine IFN-γ[25,26], whereas Eomesodermin expressing CD4⁺ T cells are necessary for the development of chronic neuroinflammation and multiple sclerosis[27,28].

Except for one study[23], all previous studies applying flow cytometric analysis in ALS patients have been restricted to peripheral blood[29–32]. In the study performed by Jin et al.[23], CSF from six ALS patients was analysed without observing significant differences between ALS patients and controls, which may be partly due to a lack of statistical power. In the present study, we made a detailed comparison in the T cell subsets of CSF and blood, to understand whether the immune profile measured in the periphery can be representative of the immune profile in the central nervous system. As the correlations of blood- and CSF-based T

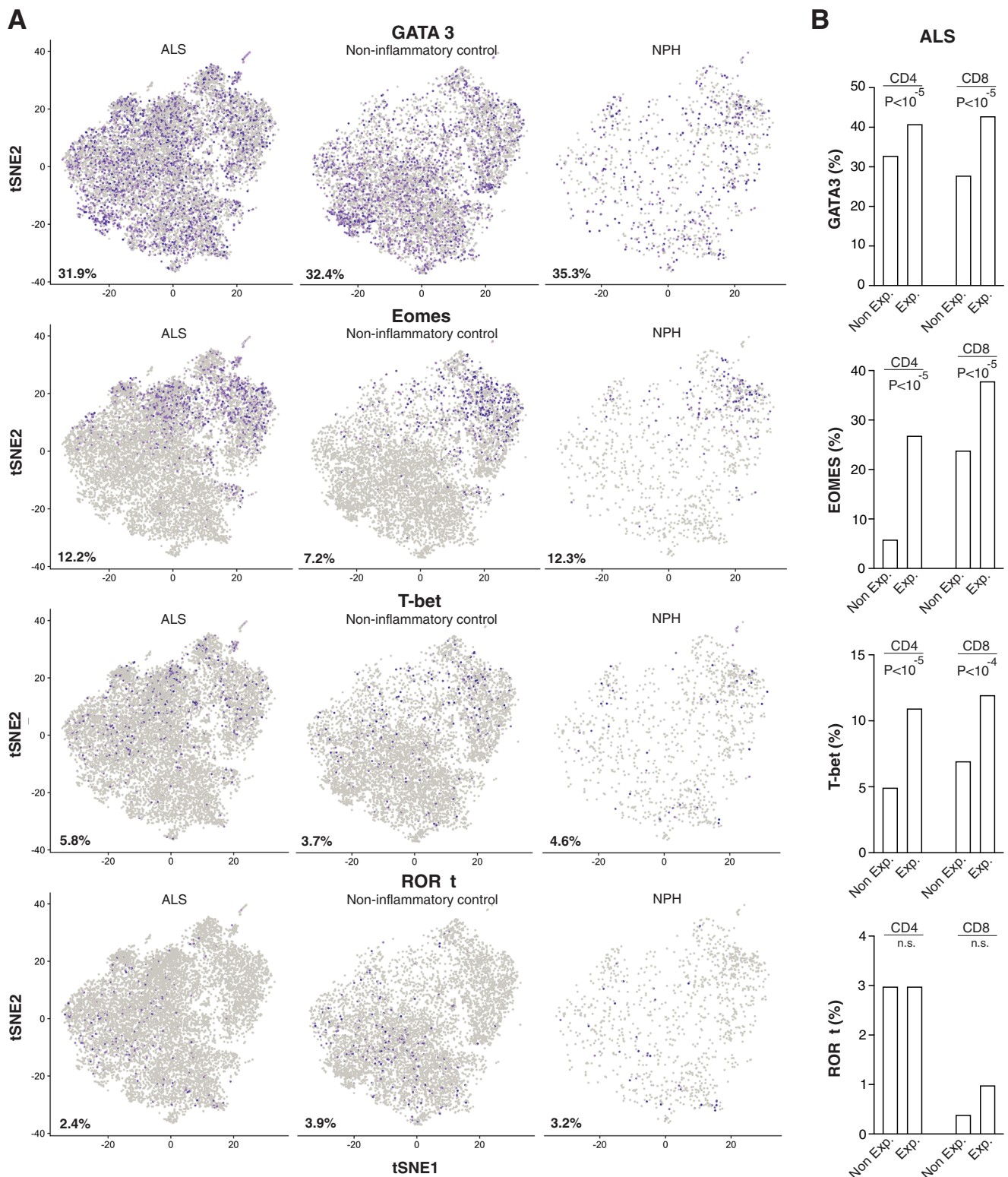

**Fig. 5 | Distinct expression of lineage-defining transcription factors among clonally expanded T cells in CSF of ALS patients.** t-SNE plots of 10X scRNA-seq data showing expression of GATA3, Eomesodermin, Tbet, and RORγt with the percentage of positive cells (shown in purple) noted (**A**). Quantification of GATA3, Eomesodermin, Tbet, and RORγt expressing cells among expanded (>5 identical TCR sequences) and non-expanded (≤5 identical TCR sequences) T cells. The *p* values were calculated using the chi-squared test (**B**).

cell markers varied between 0.20 and 0.33 and the predictive power in either direction is weak, future studies may benefit from studying both compartments to gain a more comprehensive understanding of immunological processes in ALS and other brain diseases.

The strengths of this study include a large sample with inclusion only of incident cases, the availability of both blood and CSF samples, as well as the complete follow-up for all patients through the cross-linkage to the Swedish MND Quality Registry. Although we were not

able to include all newly diagnosed ALS patients during the study period in Stockholm, mainly because of the demanding requirement of fresh sample analysis, the study sample was representative of the clinical characteristics of all ALS patients diagnosed in Stockholm during the same period. This is also the first large-scale comparison between blood and CSF that allowed us to determine to what extent blood measurements reflect the milieu in the disease target organ. In addition to studying the T cell subsets individually, we used factor analysis combined with cluster analysis to assess the synergistic effect of different T cell types and the predictive value of such T cell profiles in the potential stratification of ALS patients. Finally, we performed both flow cytometric and scRNA-seq analyses to define the compositional and functional characteristics of immune cells in the CSF of ALS patients.

Our study also has weaknesses. The flow cytometric analysis is limited to certain T cell populations. It would have been valuable to extend the analysis to all major leukocyte subsets in both blood and CSF. In addition, a larger set of samples in the scRNA-seq analysis would give better power to decipher in more detail T cell transcriptomic alterations in ALS and for ALS patients of different subtypes. Also, given the relatively low number of T cells in CSF, we were unable to perform flow cytometric analysis in parallel on the CSF samples of the scRNA-seq analysis. Future studies are needed to perform both flow cytometric analysis and scRNA-seq analysis, using, for instance, blood samples. Similarly, we had only 10 patients with *C9orf72* mutations, making it difficult to analyse these patients separately. It remains, therefore, to be examined whether T cell subsets would behave differently in *C9orf72*-related ALS. Furthermore, we have only measurement of T cell subsets at the time of ALS diagnosis. Longitudinal sampling is therefore needed to inform on potentially relevant temporal changes in the immune responses during the disease course of ALS. However, a challenge with this strategy is that ALS is a very aggressive disease and there will be a strong selection for slowly progressing disease over time. Finally, even though we included control samples in the scRNA-seq analysis, it would have been better to include a control group of smaller diversity. Similarly, although the present study is the first to describe T cell phenotypes in detail among patients with ALS, we did not include a group of controls, making it difficult to conclude if some of the phenotypes are specific to ALS.

In conclusion, this study demonstrates that the compositional and functional status of T cells in blood and CSF predicts the survival and disease progression rate of ALS. The observation of T cell clonal expansion in the intrathecal compartment represents one of the strongest arguments obtained so far for an autoimmune disease component in ALS. Together, although it is impossible to draw a firm conclusion of a causal role of any T cell subsets on ALS prognosis from the present study due to its observational design, our findings nevertheless strengthen the notion that T cell immunity is involved in modulating the disease course of ALS and may therefore constitute a therapeutic target. Functional investigations are, however still warranted to better understand the mechanisms.

## Methods

### Study population
We performed a cohort study of 89 patients with newly diagnosed ALS included between March 2016 and March 2020, and followed until October 7, 2020, at the ALS Clinical Research Centre of Karolinska University Hospital in Stockholm, Sweden. The ALS Clinical Research Centre is the only tertiary center for ALS in Stockholm, with a population of approximately 2 million inhabitants. All patients received a diagnosis of probable, possible, or definite ALS, according to the El Escorial criteria[33]. The study was approved by the Ethical Review Board in Stockholm, Sweden (DNRs 2014/1815-31/4 and 2018-1065/31). The reporting of clinical data complies with the STROBE guidelines. Oral

and written informed consent was collected from all study participants.

### Clinical data
Using the unique Swedish personal identification numbers, the participants were linked to the Swedish Motor Neuron Disease (MND) Quality Registry, which has since 2015 recruited MND patients in Sweden at the time of diagnosis with prospective collection of information on clinical measures, biological samples, and quality of life outcomes[34]. Since 2017, this registry has had 99% coverage of all MND patients in Stockholm. For all participants of the present study, we collected information on age, sex, date of symptom onset, date of diagnosis, and site of symptom onset from this registry at the time of diagnosis, whereas information on ALS Functional Rating Scale-revised (ALSFRS-R) score and body mass index (BMI) was collected both at the time of diagnosis and repeatedly at each clinic visit thereafter (approximately every three months). All patients had been screened for *C9orf72* mutations and this information was obtained from medical records review.

All study participants were followed through the MND Quality Registry from the date of diagnosis until the date of death, the date of use of invasive ventilation, or October 7, 2020, whichever occurred first. The primary study outcomes were risk of death or use of invasive ventilation and disease progression rate, which was calculated as 48 minus ALSFRS-R score divided by the time interval between the time of symptom onset and the time of ALSFRS-R measure (in months). A full score of 48 was used for the time of symptom onset. During follow-up, 50 of the 89 patients with ALS died (including two patients that started invasive ventilation) with a median survival of 510 days from diagnosis (780 days from symptom onset). During follow-up, seven patients received a final diagnosis of primary lateral sclerosis, primary spinal muscular atrophy, or other motor neuron disease, but ALS.

### Sample collection and processing
Blood and CSF samples were collected at the time of diagnosis or shortly thereafter (within 3 months after diagnosis) for all study participants. These consisted of a 3 mL blood sample collected in a sodium heparin tube (BD) and a 16 mL CSF sample collected in two 10 mL plastic tubes (Sarstedt) by lumbar puncture. Blood was kept at room temperature, while CSF was directly centrifuged (400×*g*, 10 min) with the isolated cells kept at 4 °C. Both specimens were processed fresh without intervening freezing. The average time between the sampling and the start of the experimental analysis was around two hours. Peripheral blood mononuclear cells (PBMCs) were separated using a Ficoll (GE Healthcare) density gradient centrifugation according to the manufacturer's protocol. PBMCs were washed with PBS and a total of 1 million cells were collected for further analysis. All CSF cells were washed with PBS before further analysis.

### Flow cytometry
Cells were first labeled with the Live/Dead Fixable Dead Cell Staining Kit (Life Technologies). Subsequently, the cells were incubated with a blocking reagent against the Fc receptor, followed by surface marker staining using the following antibodies (Fluorochrome, Clone, Provider, Reference, Amount used per 1 million cells): CD3 (AF488, HIT3a, BioLegend, 300320, 2 μL), CD4 (BV786, SK3, BD Biosciences, 563877, 2 μL), CD8 (BV605, SK1, BD Biosciences, 564116, 3 μL), CD127 (PE-Cy7, eBioRDR5, ThermoFisher, 25-1278-42, 0.4 μL), CD25 (PE, 3G10, ThermoFisher, MHCD2504, 1 μL), and CD45RA (APC-H7, HI100, BioLegend, 304150, 0.4 μL). Cells were then fixed and permeabilized using the eBioscience FOXP3/transcription fixation and permeabilization kit (Invitrogen) and, as a final step, stained for the following intracellular markers (Fluorochrome, Clone, Provider, Reference, Amount used per 1 million cells): FoxP3 (PE-CF594, 236 A/E7, BD Biosciences, 653955,

5 µL) and Ki67 (BV421, B56, BD Biosciences, 562899, 5 µL). Data were acquired on BD LSRFortessa (BD Biosciences) and analyzed on FlowJo 10 (BD Biosciences). Supplementary Fig. 1 is an example of the results obtained from the flow cytometric analysis. We focused on CD3$^+$, CD4$^+$, CD8$^+$, CD4$^+$FOXP3$^-$ effector T cells (Teff), Treg cells, CD25$^{++}$CD45RA$^-$ activated Treg cells (aTreg)[35], and CD25$^+$CD45RA$^+$ resting Treg cells (rTreg)[35] in blood. Because of low cell numbers and lack of a distinct CD45RA$^+$ population, we did not study aTreg and rTreg cells in CSF.

## Statistical analysis

**Primary analysis—T cell subsets and survival.** We used the Cox model to calculate the hazard ratios (HR) and their 95% confidence intervals (CI) of death "or" use of invasive ventilation in relation to the frequencies of T cell subsets, per standard deviation (SD) increase, with time since diagnosis as the underlying time scale. All models were adjusted for age at diagnosis, sex, site of onset, delay in diagnosis (time interval between symptom onset and date of diagnosis), progression rate at the time of diagnosis, BMI measured closest to diagnosis, and the time interval between BMI measurement and date of diagnosis. The proportional hazards assumption was examined based on the Schoenfeld residuals and found to hold.

**Primary analysis—T cell subsets and disease progression rate.** We used Spearman's correlation to assess the correlation of T cell subsets with disease progression rate at the time of diagnosis. Furthermore, to study the longitudinal trajectory of the ALSFRS-R score, we first plotted all ALSFRS-R measurements from diagnosis onward for all study participants. To understand whether such trajectory differed between patients with different baseline T cell subsets, we categorized the study population into three groups according to their tertile distribution of a specific cell type. Within each group, we then plotted all ALSFRS-R scores since diagnosis and used Linear mixed models with a random intercept to predict the decline of ALSFRS-R score over time in relation to the baseline frequency of a specific T cell type. Time was measured since date of diagnosis with a unit of six months. A *p*-value for the interaction term between time and different levels of a cell type was calculated, to assess whether the evolution of the ALSFRS-R score over time differed between groups.

**Secondary analysis—exploratory factor analysis (EFA) after multiple imputation.** In addition to studying the measures of flow cytometric analysis individually, we also performed an EFA to understand the synergistic effect of different measures. Because there was some degree of missingness in different measures (2–17%), we first used multiple imputation (M = 100) to impute missing values[36,37], and the mifa R package to combine the covariance matrices of the imputed data. We then used EFA to reduce the multidimensional flow cytometric data to fewer dimensions, based on the correlation structure of the cell populations of the blood samples and performed an EFA with principal component extraction and quartimax rotation[38]. We dropped the ones with low communalities (i.e., below 0.6). Between cell types with a correlation coefficient >0.95, we kept one per pair to avoid the so-called Ultra-Heywood cases, leading to a final EFA with 23 variables. Five factors were identified in this analysis. For each factor, we calculated a factor score as the unweighted sum of the T cell subsets only belonging to this specific factor. As in the primary analysis, we then used the Cox model and linear mixed model with a random intercept to assess the factor scores as predictors for survival and disease progression rate, respectively.

**Secondary analysis—cluster analysis.** Finally, to assess whether the studied T cell subsets could help to stratify unique patient groups, we performed a cluster analysis by using k-means clustering. The aim of the cluster analysis was to group similar observations, as opposed to factor analysis, where we grouped variables. The obtained cluster solution with four clusters presented was selected based on its predictive value for disease progression.

## Single-cell RNA sequencing (scRNA-seq)

To gain a better understanding of the functional status of T cells in ALS, we additionally performed 5' scRNA-seq in a separate group of five patients with newly diagnosed ALS and four controls using the 10x Genomics platform on isolated CSF cells. CSF samples collected from controls included two patients under investigation for normal pressure hydrocephalus (NPH) and two non-inflammatory controls, including one patient with cervical radiculopathy and one healthy control. Controls were recruited from the StopMS (Stockholm Prospective Assessment of Multiple Sclerosis) study[39], which was approved by the Ethical Review Board in Stockholm, Sweden (DNRs 2009/2107-31/2 and 2021-02060).

Using this data, we compared the composition of different leukocytes, as well as gene expression and clonal expansion of T cells, between ALS patients and controls. Details of sample preparation, sequencing, and data analysis are provided in Supplementary Methods.

## Reporting summary

Further information on research design is available in the Nature Portfolio Reporting Summary linked to this article.

## Data availability

Due to ethical and legal reasons, the flow cytometric and clinical data of ALS patients are not publicly available. Such data might, however, be shared on reasonable request to the corresponding authors. The RNAseq data and a description of the data have been deposited at https://ega-archive.org/studies/EGAS00001006675.

## Code availability

All custom codes used for data preparation and primary analysis are deposited on GitHub (https://github.com/eudoraleer/als_scrnaseq)[40].

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

## Acknowledgements

The study was supported by the European Research Council (ERC) Starting Grant (MegaALS, No. 802091 to F.F.), the Swedish Research Council (No. 2019-01088 to F.F.), Ulla-Carin Lindqvist's Foundation (to C.I. and J.A.), Bjorklunds fund (to C.I.), Neuro Sweden (to C.I.), Konung Gustaf V:s och Drottning Victorias Frimurarestiftelse (to C.I.), and the Karolinska Institutet (Senior Researcher Award and Strategic Research Area in Epidemiology to F.F.). The authors acknowledge all ALS patients and their families participating in ongoing research at the ALS Clinical Research Centre at the Karolinska University Hospital, and the Eukaryotic Single Cell Genomics (ESCG) facility in Stockholm funded by Science for Life Laboratory, KI Core, and StratRegen. The computation and data handling were enabled by resources provided by the Swedish National Infrastructure for Computing (SNIC), partially funded by the Swedish Research Council (No. 2018-05973).

## Author contributions

All authors provided critical feedback and helped shape the research, analysis, and final manuscript. S.Y., C.S., and C.C carried out the experiments and contributed to the data analysis; A.L. and L.P. contributed to the data analysis; Y.P., T.N.V., and L.Y. supervised the data analysis; L.S.W., A.-L.J., and J.A. supervised the experimental analysis; F.P., U.K., R.P., K.S., B.E., and C.I. contributed to the sample collection; A.-L.J. contributed to the development of the research protocol; C.I., J.A., and F.F. developed the idea and supervised the project.

## Funding

## Competing interests

The authors declare no competing interests.
