## [Peer Review File · Nature Communications]

T cell responses at diagnosis of amyotrophic lateral sclerosis predict disease progressionREVIEWER COMMENTS

Reviewer #1 (Remarks to the Author):

I have reviewed the paper "T cell responses at diagnosis of amyotrophic lateral sclerosis predict disease progression" by Yadzani et al.

The paper has several points of strength. The topic of the paper is of considerable interest, and little information is currently available to address a potential immunological mechanism associated with ALS. The relatively large cohort is another point of strength.

The study has moderate novelty, since potential T reg alterations in ALS have been pointed out in several previous studies.

The study is associated with two main weaknesses.

First, the data is presented in a highly summarized form, and the primary data showing the actual number of different subsets at different time points, in histogram formats, with appropriate statistics and correction for multiple comparison need to be presented. Also data from control matched cohort need to be presented.

Second, the paper is descriptive and does not establish causality, and there is no information regarding the immune alterations reported in terms of antigen specificity or functional assays to establish the functional relevance of the observations. As such the insights derived from the analysis are rather limited.

Reviewer #2 (Remarks to the Author):

This is an interesting paper describing T cell subsets in a cohort of ALS patients following by single cell sequencing to characterise the T cell expression profiles in ALS.

The initial description of the T cells (pages 4-5) states that removing ALS mimics or C9orf72-ALS cases changed the results marginally. However, I would have expected ALS mimics to have been removed from the outset, as levels of Tregs have not been associated with disease prognosis (as far as I am aware). Also, is there any evidence in the literature of C9orf72-ALS being any different in Treg response? Looking at these as a separate group may be interesting, depending on numbers?

In relation to patient numbers, whilst the abstract states this work was done on a cohort of 89 patients, in actual fact, the data varies from 62-73 patients in the blood and 67-80 in the CSF – so data was not available from all patients included in the cohort? What would be the number of patients on which all data was available? Would this not be more consistent, with the additional samples included within the supplementary data?

Five ALS cases and 4 controls (3 disease and one healthy control) were used for scRNA-seq. Were baseline Treg levels available for these cases? Did they show low, medium and high levels as described in the earlier categorisation? It would be interesting to comment in the discussion that scRNA-seq of different subtypes of patients would be interesting. There is also a wide variability in the control group, which is likely to impact significantly in the differential expression analysis.

The discussion concludes that modulating T cells may constitute a therapeutic target for ALS. However, there is no reference to the IL-2 treatment strategy (Camu et al, 2020) or T-reg infusions (Thonhoff et al, 2018) clinical trials.

Minor comments:

In using an abbreviation for the SOD1 G93A mouse model for ALS, I would recommend using SOD1, rather than just SOD.

Reviewer #3 (Remarks to the Author):

The manuscript by Fang et al presents the results of a study of 89 newly diagnosed ALS patients and reports that high frequency of CD4+FoxP3- T cells was associated with faster disease

progression and poorer survival whereas high frequency of activated Tregs and a high ratio of activated to resting Tregs in blood was associated with slower disease progression and better survival. In CSF association of Teff cells with survival showed a trend but was not statistically significant. However, the association of T cell subsets with disease progression was stronger in CSF than in blood. An RNA-seq analysis of the CSF of 5 ALS pts demonstrated increased activated CD4+ cytotoxic lymphs and decreased monocytes compared to 4 controls. Further the ALS CSF had evidence of increased T cell expansion especially of CD4+ and CD8 T cells with an activated phenotype; the expanded T cells had distinctive transcription factors compared to non-expanded T cells.

Overall this study provides a significant phenotypic characterization of Teff and Tregs and their association with survival and disease progression in ALS in a Swedish cohort.

Many previous reports have suggested that increased Teff and decreased Treg are associated with enhanced disease progression and survival, but the present report is far more definitive and also more meaningful, especially in using both blood and CSF in ALS patients at time of diagnosis to predict disease progression.

Clarification is needed with respect to the use of "activated" in describing T cells state. For example Treg is referred to as "activated or "resting." Yet the the designation is unclear. Are "activated" Tregs characterized as CCL5, CD45A-, and FoXP3++; are resting CCR7, CD45RA+ and FoXP3 low ???. Fig 2A Factor5-top right would suggest that FoxP3rTreg and FoxP3a Treg have relatively equivalent FoxP3 levels, so the definition of "activated" must not be dependent on FoxP3 levels. All designations of "activated" whether for Treg, CD4+ or CD8+ must be clarified. The values documenting a higher risk of death in the Supplementary Tables are extremely important and should not be relegated to Supplementary Tables. The specific data can be incorporated into the main text at line 87. line 90, and line 93.

REVIEWER

COMMENTS

Reviewer #1 (Remarks to the Author):

I have reviewed the paper “T cell responses at diagnosis of amyotrophic lateral sclerosis predict disease progression” by Yadzani et al.

The paper has several points of strength. The topic of the paper is of considerable interest, and little information is currently available to address a potential immunological mechanism associated with ALS. The relatively large cohort is another point of strength.

Response:

Thank you for the encouraging comments.

The study has moderate novelty, since potential T regs alterations in ALS have been pointed out in several previous studies.

Response:

We agree with the reviewer that Tregs have been studied previously in ALS. Precisely because of the promising results shown in previous studies, we performed the present study to further our understanding of Tregs and their roles in modulating disease progression of ALS.

In this study, we have for the first time studied the repertoire of T cells in the peripheral blood and the intracranial compartment simultaneously and made the following novel discoveries:

1. CD4⁺FOXP3⁻ effector T cells are indicative of survival and disease progression rate of ALS.
2. The previously suggested protective role of Treg cells is likely attributable to the activation of Tregs.
3. Composition of T cells in blood at the time of ALS diagnosis predicts the disease course of ALS.
4. Composition of T cells in blood only partly reflects the composition of T cells in cerebrospinal fluid (CSF).
5. There is clonal expansion of effector T cells in the CSF of ALS patients.

We have clarified the added values of the present study in the manuscript (Discussion, page 09, paragraph 02):

“The present study is to our knowledge the first to simultaneously define the immunophenotype in both the peripheral and intrathecal compartment at the time of ALS diagnosis, and to relate such phenotype to disease progression. We found that a high frequency of effector T cells was indicative of a poor survival, whereas a high frequency of activated Treg (or a high ratio of activated to resting Treg) cells was indicative of a better survival, after ALS diagnosis. T cell subsets measured in both blood and CSF were also predictive of disease progression rate after ALS diagnosis. Finally, CSF T cells exhibited differential gene expression and clonal expansion patterns between ALS patients and individuals free of ALS.”

The study is associated with two main weaknesses.

Response:

Thank you for the comment. We have now addressed the weaknesses below and made changes to the manuscript accordingly.

First, the data is presented in a highly summarized form, and the primary data showing the actual number of different subsets at different time points, in histogram formats, with appropriate statistics and correction for multiple comparison need to be presented. Also data from control matched cohort need to be presented.

Response:

Thank you for the suggestion. We have now added the primary data showing the proportions of different subsets, both in histogram formats and as a table, in the Supplementary Figure S1 (see below). We have also added comment on these new results in the manuscript (Results, page 04, paragraph 04):

*“Flow cytometry was used to define T cell subsets in blood and CSF samples collected from each patient at the time of diagnosis, to identify immune markers associated with ALS survival. **Supplementary Figure S1** shows the gating strategy and summary of primary data for T cell subsets.”*

Figure S1

Figure S1. Gating strategy and summary of primary data for T cell subsets. Flow cytometry was used to define distinct T cell subsets. Gating strategy and distribution of T cell subsets in blood (A) and cerebrospinal fluid (B). Mean and standard deviation (SD) of frequencies of T cell subsets (C).

As we used T cell subsets measured at the time of ALS diagnosis, we could not provide data across different time points. Accordingly, we have not applied any statistical test and believe that there is no need to correct for multiple testing. We have clarified this in the manuscript (Discussion, page 13, paragraph 01):

“Furthermore, we had only measurement of T cell subsets at the time of ALS diagnosis. Longitudinal sampling is therefore needed to inform on potentially relevant temporal changes in the immune responses during the disease course of ALS. However, a challenge with this strategy is that ALS is a very aggressive disease and there will be a strong selection for slowly progressing disease over time.”

Finally, we also agree with the reviewer that it would have been great to include controls in this study. However, as we did not have the ethical approval to sample CSF from healthy controls in the present study (apart from the healthy controls of the scRNA-seq analysis that were recruited from the StopMS study), we do not have access to such data. Instead of comparing T cell subsets between ALS patients and healthy controls, we designed the study to mainly understand the role of different T cell subsets in periphery and intracranial compartment on the survival and disease progression rate of ALS. We have now added comments in the manuscript to discuss the need of future studies to contrast ALS patients and ALS-free individuals (Discussion, page 13, paragraph 01):

“Similarly, although the present study is the first to describe T cell phenotypes in detail among patients with ALS, we did not include a group of controls, making it difficult to conclude if some of the phenotypes are specific to ALS.”

Second, the paper is descriptive and does not establish causality, and there is no information regarding the immune alterations reported in terms of antigen specificity or functional assays to establish the functional relevance of the observations. As such the insights derived from the analysis are rather limited.

Response:

We agree with the reviewer that it is difficult to establish causality in the present study due to its observational nature. Indeed, it is difficult to establish the causal role of immune cells on ALS in any human observational studies, because of the complexity of immune responses in general and the potentially dual role of immune responses in ALS (Murdock et al. 2005), as well as the presumably long pre-clinical stage of ALS. We further agree with the reviewer that it would have been very interesting to perform functional investigations following the findings of the present study. We are currently trying to identify the T cell epitopes that induce CD4⁺ CTL expansion among ALS patients. We have now commented on this in the manuscript (Discussion, page 13, paragraph 02):

“Together, although it is impossible to draw a firm conclusion of a causal role of any T cell subsets on ALS prognosis from the present study due to its observational design, our findings nevertheless strengthen the notion that T cell immunity is involved in modulating the disease course of ALS. Functional investigations are however still warranted to better understand the mechanisms.”

Reviewer #2 (Remarks to the Author):

This is an interesting paper describing T cell subsets in a cohort of ALS patients following by single cell sequencing to characterise the T cell expression profiles in ALS.

Response:

Thank you for the positive comment.

The initial description of the T cells (pages 4-5) states that removing ALS mimics or C9orf72-ALS cases changed the results marginally. However, I would have expected ALS mimics to have been removed from the outset, as levels of Tregs have not been associated with disease prognosis (as far as

I am aware). Also, is there any evidence in the literature of C9orf72-ALS being any different in Treg response? Looking at these as a separate group may be interesting, depending on numbers?

Response:

Thank you for the comments. In the group of “ALS mimics”, we included patients with a diagnosis of motor neuron disease (MND), primary lateral sclerosis (PLS), or primary spinal muscular atrophy (PSMA), which may all be included in ALS spectrum. We decided therefore in the original manuscript to *not* exclude these patients (N=7) from the cohort. We have now clarified this in the manuscript (Results, page 05, paragraph 01):

“Exclusion of patients with ALS mimic diseases including primary lateral sclerosis, primary spinal muscular atrophy, and other motor neuron disease (N=7) (Supplementary Table 2) or patients with C9orf72 mutations (N=10) (Supplementary Table 3) only changed these results marginally.”

To our best knowledge, no study has examined the role of Treg cells on disease progression of these mimics. The below tables and the Supplementary Table 2 of the original manuscript show that the characteristics of patient cohort do not differ greatly with or without exclusion of these mimics (Table R1) and the key results remained largely unchanged after exclusion of these mimics from the analysis (Table R2). Although we would prefer to keep the main analysis as is, we are willing to reconsider our position if the editor and reviewer would disagree with us.

Table R1. Characteristics of the study cohort with and without exclusion of ALS mimics, according to the Swedish Motor Neuron Disease Quality Registry.

Characteristics	Entire study cohort (n=89)	After exclusion of ALS mimics (n=82)
Age at diagnosis, years		
Mean (SD)	66.52 (10.69)	66.28 (10.84)
Sex, N (%)		
Male	54 (60.67%)	48 (58.54%)
Female	35 (39.33%)	34 (41.46%)
Final diagnosis, N (%)		
ALS	82 (92.13%)	82 (100%)
Other MND	7 (7.87%)	
Site of onset, N (%)		
Bulbar	38 (42.70%)	38 (46.34%)
Non-bulbar	51 (57.30%)	44 (53.66%)
Other	-	
ALSFRRS-R score at diagnosis, mean (SD)	38.29 (7.85)	37.99 (8.06)
Progression rate at diagnosis, mean (SD)	0.81 (0.82)	0.84 (0.84)
Diagnostic delay, median	377 days	382 days
Survival status at end of follow-up, N (%)		
Dead	50 (56.18%)	49 (59.76%)
Alive	39 (43.82%)	33 (40.24%)

Table R2. Hazard ratios (HRs) and their 95% confidence intervals (CIs) of death (or use of invasive ventilation) in relation to frequency of T cell subsets in blood or cerebrospinal fluid (CSF), after adjustment for age at diagnosis, sex, site of onset, diagnostic delay, disease progression rate at diagnosis, body mass index (BMI), and time difference between measurement of BMI and blood sampling, analysis using the entire study cohort and after exclusion of ALS mimics.

Cell population out of live cells (%)	Tertile	Entire study cohort		After exclusion of ALS mimics	
		No. of patients (outcomes)	HR (95% CI)	No. of patients (outcomes)	HR (95% CI)
Blood					
CD3+	Low	26 (12)	Ref	25 (13)	Ref
	Medium	26 (16)	2.47 (1.11-5.52)	24 (16)	2.82 (1.28-6.20)
	High	26 (16)	2.49 (1.03-6.03)	24 (15)	2.54 (1.08-5.92)
CD4+	Low	26 (13)	Ref	25 (14)	Ref
	Medium	26 (15)	1.82 (0.81-4.06)	24 (14)	2.40 (1.05-5.46)
	High	25 (16)	2.29 (1.04-5.04)	24 (16)	3.11 (1.38-7.01)
CD8+	Low	22 (13)	Ref	21 (14)	Ref
	Medium	22 (13)	1.04 (0.44-2.46)	21 (12)	1.29 (0.53-3.13)
	High	22 (09)	0.81 (0.32-2.06)	20 (09)	0.91 (0.35-2.32)
Teff	Low	26 (13)	Ref	24 (13)	Ref
	Medium	26 (14)	1.61 (0.72-3.62)	24 (15)	2.79 (1.22-6.41)
	High	25 (17)	2.43 (1.10-5.37)	24 (16)	2.80 (1.23-6.39)
Treg	Low	26 (12)	Ref	24 (13)	Ref
	Medium	26 (14)	1.44 (0.64-3.23)	24 (13)	1.15 (0.50-2.64)
	High	25 (18)	1.39 (0.64-3.00)	24 (18)	1.89 (0.83-4.29)
aTreg	Low	26 (15)	Ref	24 (15)	Ref
	Medium	25 (12)	0.40 (0.17-0.92)	24 (12)	0.38 (0.16-0.89)
	High	25 (17)	0.76 (0.37-1.59)	23 (17)	0.99 (0.47-2.09)
rTreg	Low	26 (13)	Ref	24 (14)	Ref
	Medium	26 (16)	1.65 (0.76-3.60)	24 (15)	1.76 (0.76-4.08)
	High	25 (15)	1.59 (0.69-3.66)	24 (15)	1.91 (0.82-4.47)
aTreg/rTreg (ratio)	Low	26 (14)	Ref	24 (13)	Ref
	Medium	25 (16)	1.06 (0.46-2.43)	24 (17)	1.20 (0.52-2.78)
	High	25 (14)	0.56 (0.24-1.31)	23 (14)	0.68 (0.30-1.57)
CSF					
CD3+	Low	27 (15)	Ref	25 (15)	Ref
	Medium	27 (17)	2.21 (1.01-4.82)	25 (16)	1.89 (0.84-4.23)
	High	26 (14)	1.24 (0.54-2.82)	25 (15)	1.60 (0.69-3.71)
CD4+	Low	27 (14)	Ref	25 (14)	Ref
	Medium	27 (16)	1.79 (0.82-3.93)	25 (16)	2.18 (1.00-4.78)
	High	26 (16)	3.04 (1.24-7.46)	25 (16)	3.19 (1.31-7.79)
CD8+	Low	23 (10)	Ref	21 (10)	Ref
	Medium	22 (16)	1.22 (0.47-3.15)	21 (15)	1.03 (0.39-2.71)
	High	22 (11)	0.78 (0.30-2.06)	21 (12)	1.01 (0.40-2.56)
Teff	Low	27 (14)	Ref	25 (14)	Ref
	Medium	26 (15)	1.68 (0.75-3.77)	25 (16)	2.16 (0.98-4.75)
	High	26 (17)	3.18 (1.33-7.64)	24 (16)	3.22 (1.32-7.83)
Treg	Low	27 (17)	Ref	25 (17)	Ref
	Medium	26 (13)	0.96 (0.44-2.08)	25 (12)	0.92 (0.41-2.06)
	High	26 (16)	0.87 (0.41-1.86)	14 (17)	1.62 (0.69-3.78)

We are grateful for the very pertinent suggestion to study *C9orf72* patients separately. However, as correctly assumed by the reviewer, we have a limited number of *C9orf72* patients in this study (n=10), making it difficult to perform separate analysis. We have added comments regarding this in the manuscript (Discussion, page 13, paragraph 01):

“Similarly, we had only 10 patients with C9orf72 mutations, making it difficult to analyze these patients separately. It remains therefore to be examined whether T cell subsets would behave differently in C9orf72-related ALS.”

In relation to patient numbers, whilst the abstract states this work was done on a cohort of 89 patients, in actual fact, the data varies from 62-73 patients in the blood and 67-80 in the CSF – so data was not available from all patients included in the cohort? What would be the number of patients on which all data was available? Would this not be more consistent, with the additional samples included within the supplementary data?

Response:

Thank you for pointing this out. The number of patients varied between different analyses because of the following reasons.

First, among the total 89 patients, four patients had missing data on BMI whereas another four patients had missing data on ALSFRS-R score. BMI and ALSFRS-R score were controlled for as covariates in the survival analysis, but not in other analyses.

Moreover, not all patients had available data on T cell subsets in blood and/or CSF. This is due to either an unsuccessful flow cytometric analysis or having access to only one of the specimens.

As suggested by the reviewer, we have now performed analyses including only patients with complete data (N=63) (please see the following tables).

We have explained this and added these new results in the manuscript (Results, page 05, paragraph 03), including two new supplementary tables.

*“Although the entire study cohort included 89 patients, not all patients were included in all analyses, due to missing data on covariables (i.e., BMI and ALSFRS-R score) or T cell subsets (i.e., lack of specimen or unsuccessful flow cytometric analysis). A sensitivity analysis including only patients with complete data (N=63) showed similar patient characteristics (**Supplementary Table 4**) and results (**Supplementary Table 5**).”*

Supplementary Table 4. Characteristics of the entire study cohort and the cohort including only patients with no missing data, according to the Swedish Motor Neuron Disease Quality Registry.

Characteristics	Entire study cohort (n=89)	After exclusion of patients with missing data (n=63)
Age at diagnosis, years		
Mean (SD)	66.52 (10.69)	66.32 (11.68)
Sex, N (%)		
Male	54 (60.67%)	39 (61.90%)
Female	35 (39.33%)	24 (38.10%)
Final diagnosis, N (%)		
ALS	82 (92.13%)	59 (93.65%)
Other MND	7 (7.87%)	4 (6.35%)
Site of onset, N (%)		
Bulbar	38 (42.70%)	28 (44.44%)
Non-bulbar	51 (57.30%)	35 (55.56%)
Other	-	
ALSFRS-R score at diagnosis, mean (SD)	38.29 (7.85)	38.20 (8.11)
Progression rate at diagnosis, mean (SD)	0.81 (0.82)	0.80 (0.77)
Diagnostic delay, median	377 days	376 days
Survival status at end of follow-up, N (%)		
Dead	50 (56.18%)	33 (52.38%)
Alive	39 (43.82%)	30 (47.62%)

Supplementary Table 5. Hazard ratios (HRs) and their 95% confidence intervals (CIs) of death (or use of invasive ventilation) in relation to frequency of T cell subsets in blood or cerebrospinal fluid (CSF), after adjustment for age at diagnosis, sex, site of onset, diagnostic delay, disease progression rate at diagnosis, body mass index (BMI), and time difference between measurement of BMI and blood sampling, analysis using the entire study cohort and after exclusion of patients with missing data.

Cell population out of live cells (%)	Tertile	Entire study cohort (n=89)		After exclusion of patients with missing data (n=63)	
		No. of patients (outcomes)	HR (95% CI)	No. of patients (outcomes)	HR (95% CI)
Blood					
CD3+	Low	26 (12)	Ref	21 (11)	Ref
	Medium	26 (16)	2.47 (1.11-5.52)	21 (12)	2.41 (0.92-6.36)
	High	26 (16)	2.49 (1.03-6.03)	21 (12)	1.84 (0.70-4.81)
CD4+	Low	26 (13)	Ref	21 (10)	Ref
	Medium	26 (15)	1.82 (0.81-4.06)	21 (15)	2.92 (1.25-6.79)
	High	25 (16)	2.29 (1.04-5.04)	21 (10)	1.71 (0.66-4.41)
CD8+	Low	22 (13)	Ref	21 (13)	Ref
	Medium	22 (13)	1.04 (0.44-2.46)	21 (13)	1.11 (0.48-2.57)
	High	22 (09)	0.81 (0.32-2.06)	21 (9)	0.74 (0.29-1.86)
Teff	Low	26 (13)	Ref	21 (11)	Ref
	Medium	26 (14)	1.61 (0.72-3.62)	21 (13)	2.06 (0.88-4.83)
	High	25 (17)	2.43 (1.10-5.37)	21 (11)	1.79 (0.72-4.47)
Treg	Low	26 (12)	Ref	21 (10)	Ref
	Medium	26 (14)	1.44 (0.64-3.23)	21 (11)	1.65 (0.67-4.10)
	High	25 (18)	1.39 (0.64-3.00)	21 (14)	1.28 (0.55-2.97)
aTreg	Low	26 (15)	Ref	21 (11)	Ref
	Medium	25 (12)	0.40 (0.17-0.92)	21 (09)	0.55 (0.22-1.42)
	High	25 (17)	0.76 (0.37-1.59)	21 (15)	1.03 (0.45-2.35)
rTreg	Low	26 (13)	Ref	21 (12)	Ref
	Medium	26 (16)	1.65 (0.76-3.60)	21 (11)	1.11 (0.44-2.80)
	High	25 (15)	1.59 (0.69-3.66)	21 (12)	1.89 (0.73-4.92)
aTreg/rTreg (ratio)	Low	26 (14)	Ref	21 (11)	Ref
	Medium	25 (16)	1.06 (0.46-2.43)	21 (12)	0.89 (0.34-2.30)
	High	25 (14)	0.56 (0.24-1.31)	21 (12)	0.40 (0.15-1.06)
CSF					
CD3+	Low	27 (15)	Ref	21 (11)	Ref
	Medium	27 (17)	2.21 (1.01-4.82)	21 (14)	3.22 (1.21-8.57)
	High	26 (14)	1.24 (0.54-2.82)	21 (10)	1.36 (0.50-3.65)
CD4+	Low	27 (14)	Ref	21 (12)	Ref
	Medium	27 (16)	1.79 (0.82-3.93)	21 (11)	1.57 (0.64-3.83)
	High	26 (16)	3.04 (1.24-7.46)	21 (12)	3.30 (1.08-10.1)
CD8+	Low	23 (10)	Ref	21 (09)	Ref
	Medium	22 (16)	1.22 (0.47-3.15)	21 (14)	1.32 (0.48-3.61)
	High	22 (11)	0.78 (0.30-2.06)	21 (12)	1.06 (0.39-2.89)
Teff	Low	27 (14)	Ref	21 (12)	Ref
	Medium	26 (15)	1.68 (0.75-3.77)	21 (11)	1.88 (0.76-4.67)
	High	26 (17)	3.18 (1.33-7.64)	21 (12)	2.06 (0.76-5.58)
Treg	Low	27 (17)	Ref	21 (14)	Ref
	Medium	26 (13)	0.96 (0.44-2.08)	21 (08)	0.72 (0.29-1.81)
	High	26 (16)	0.87 (0.41-1.86)	21 (13)	0.93 (0.40-2.18)

Five ALS cases and 4 controls (3 disease and one healthy control) were used for scRNA-seq. Were baseline Treg levels available for these cases? Did they show low, medium and high levels as described in the earlier categorisation? It would be interesting to comment in the discussion that scRNA-seq of different subtypes of patients would be interesting. There is also a wide variability in the control group, which is likely to impact significantly in the differential expression analysis.

Response:

Thank you for this very relevant comment. Indeed, we had appreciated the opportunity to run flow cytometric analysis on CSF samples in parallel to scRNA-seq. However, due to the low concentrations of T cells in CSF and a limited volume that can be sampled, this would have jeopardized the outcome of the experiment. We have now clarified this in the manuscript, and call for studies to perform both flow cytometric analysis and scRNA-seq using blood samples (Discussion, page 13, paragraph 01):

“Also, given the relatively low number of T cells in CSF, we were unable to perform flow cytometric analysis in parallel on the CSF samples of the scRNA-seq analysis. Future studies are needed to perform both flow cytometric analysis and scRNA-seq analysis, using for instance blood samples.”

We also agree with the reviewer that it is highly interesting to study scRNA-seq data for ALS patients of different subtypes. In this study we had CSF samples from 2 patients with a slow-progressing disease and 3 patients with a fast-progressing disease, according to disease progression rate estimated at the time of diagnosis. Larger studies are therefore needed in the future in this regard. We have now commented this in the manuscript (Discussion, page 13, paragraph 01):

“In addition, a larger set of samples in the scRNA-seq analysis would give better power to decipher in more detail T cell transcriptomic alterations in ALS and for ALS patients of different subtypes.”

Finally, we also agree with the reviewer that our control group of the scRNA-seq analysis is not perfect. This was partly due to feasibility as we recruited the controls from the StopMS study due to a lack of ethical approval to sample CSF from healthy controls in the ALS study. We have now discussed this limitation in the manuscript (Discussion, page 13, paragraph 01):

“Finally, although we included control samples in the scRNA-seq analysis, it would have been better to include a control group of smaller diversity.”

The discussion concludes that modulating T cells may constitute a therapeutic target for ALS. However, there is no reference to the IL-2 treatment strategy (Camu et al, 2020) or T-reg infusions (Thonhoff et al, 2018) clinical trials.

Response:

We apologize for not citing these important studies and have now added them in the manuscript (Introduction, page 04, paragraph 01).

“Clinical trials involving therapeutic targeting of T cells in ALS are still rare. However, in a phase 2 randomized trial, the administration of low-dose interleukin-2 (ld-IL-2) was shown to be well tolerated and increase the percentage of Treg cells.¹⁴ In another study, infusion of expanded autologous Tregs was found to lead to an increasing percentage and suppressive function of Tregs and slowing of the disease progression.²”

Minor comments:

In using an abbreviation for the SOD1 G93A mouse model for ALS, I would recommend using SOD1, rather than just SOD.

Response:

Thank you very much for pointing this out. We have corrected the abbreviations accordingly.

Reviewer #3 (Remarks to the Author):

The manuscript by Fang et al presents the results of a study of 89 newly diagnosed ALS patients and reports that high frequency of CD4⁺FoxP3⁻ T cells was associated with faster disease progression and poorer survival whereas high frequency of activated Tregs and a high ratio of activated to resting Tregs in blood was associated with slower disease progression and better survival. In CSF association of Teff cells with survival showed a trend but was not statistically significant. However, the association of T cell subsets with disease progression was stronger in CSF than in blood. An RNA-seq analysis of the CSF of 5 ALS pts demonstrated increased activated CD4⁺ cytotoxic lymphs and decreased monocytes compared to 4 controls. Further the ALS CSF had evidence of increased T cell expansion especially of CD4⁺ and CD8 T cells with an activated phenotype; the expanded T cells had distinctive transcription factors compared to non-expanded T cells.

Overall this study provides a significant phenotypic characterization of Teff and Tregs and their association with survival and disease progression in ALS in a Swedish cohort.

Many previous reports have suggested that increased Teff and decreased Treg are associated with enhanced disease progression and survival, but the present report is far more definitive and also more meaningful, especially in using both blood and CSF in ALS patients at time of diagnosis to predict disease progression.

Response:

Thank you very much for the very positive comments!

Clarification is needed with respect to the use of "activated" in describing T cells state. For example Treg is referred to as "activated" or "resting." Yet the the designation is unclear. Are "activated" Tregs characterized as CCL5, CD45A⁻, and FoxP3⁺⁺; are resting CCR7, CD45RA⁺ and FoxP3 low ?? Fig 2A Factor5-top right would suggest that FoxP3^rTreg and FoxP3^a Treg have relatively equivalent FoxP3 levels, so the definition of "activated" must not be dependent on FoxP3 levels. All designations of "activated" whether for Treg, CD4⁺ or CD8⁺ must be clarified.

Response:

We apologize for not clarifying these in the original manuscript. Activated Treg cells are defined as CD25⁺⁺CD45RA⁻ whereas resting Treg cells are defined as CD25⁺CD45RA⁺. Please see our gating strategy shown in Supplementary Figure S1, which follows the gating strategies used in our previous study (Andersson, J. 2007) and the study of Miyara et al. (Miyara, M. 2009). We have now clarified this in the manuscript (Methods, page 16, paragraph 02).

"We focused on CD3⁺, CD4⁺, CD8⁺, CD4⁺FOXP3⁻ effector T cells (Teff), Treg cells, CD25⁺⁺CD45RA⁻ activated Treg cells (aTreg)³⁴, and CD25⁺CD45RA⁺ resting Treg cells (rTreg)³⁴ in blood."

Furthermore, resting T cells and activated T cells were defined using CCR7 and CCL5 markers, respectively. This was shown in Figure S3 and we have now clarified the activation markers in the manuscript as well (Results, page 07, paragraph 02 and Results, page 08, paragraph 02).

"CSF cells of ALS patients displayed increased amounts of CD4⁺ cytotoxic lymphocytes (CTLs) and CD4⁺ T cells with an activated phenotype (CCR7⁺CCL5⁺)¹⁵ but decreased amounts of monocytes and CD4⁺CD8⁺ double positive T cells, compared with controls."

"ALS patients exhibited greater levels of TCR expansions in CD4⁺ CTLs as well as in CD4⁺ and CD8⁺ T cells with an activated phenotype (CCR7⁺CCL5⁺), compared with controls (Figure 4A, B)."

The values documenting a higher risk of death in the Supplementary Tables are extremely important and should not be relegated to Supplementary Tables. The specific data can be incorporated into the main text at line 87. line 90, and line 93.

Response:

Thank you for the positive comments. We have now moved Supplementary Table 4 into the main text as a new Table 2, as suggested by the reviewer.

REVIEWERS' COMMENTS

Reviewer #1 (Remarks to the Author):

I believe the authors satisfactorily addressed my concerns, and the manuscript is much improved.

Reviewer #2 (Remarks to the Author):

You have addressed each of the comments made by this reviewer appropriately.

With regard to the ALS mimics, I am happy for these to be included and appreciate the clearer justification of what these mimics were. Would it be better to describe them as "ALS spectrum disorders" rather than mimics, which I would view as diseases such as multiple system atrophy, SCA3 and Kennedy's disease?

Reviewer #3 (Remarks to the Author):

All concerns have been adequately addressed.

REVIEWERS' COMMENTS

Reviewer #1 (Remarks to the Author):

I believe the authors satisfactorily addressed my concerns, and the manuscript is much improved.

Reply: Thank you for your positive comment.

Reviewer #2 (Remarks to the Author):

You have addressed each of the comments made by this reviewer appropriately.

With regard to the ALS mimics, I am happy for these to be included and appreciate the clearer justification of what these mimics were. Would it be better to describe them as "ALS spectrum disorders" rather than mimics, which I would view as diseases such as multiple system atrophy, SCA3 and Kennedy's disease?

Reply: Thank you for your positive comment. We have now changed the wording "mimics" to "ALS spectrum disorders", as suggested.

Reviewer #3 (Remarks to the Author):

All concerns have been adequately addressed.

Reply: Thank you for your positive comment.